# Biomechanics of running: A special reference to the comparisons of wearing boots and running shoes

**Alireza Shamsoddini**[1]*, **Mohammad Taghi Hollisaz**[2]

**1** Exercise Physiology Research Center, LifeStyle Institute, Baqiyatallah University of Medical Sciences, Tehran, Iran, **2** Department of physical Medicine and Rehabilitation, Baqiyatallah University of Medical Sciences, Tehran, Iran

* alirezaot@bmsu.ac.ir

**Data Availability Statement:** All relevant data are within the paper and its Supporting Information files.

**Funding:** The author(s) received no specific funding for this work.

## Abstract

Boots are often used in sports, occupations, and rehabilitation. However, there are few studies on the biomechanical alterations after wearing boots. The current study aimed to compare the effects of running shoes and boots on running biomechanics. Kinematics and ground reaction forces were recorded from 17 healthy males during running at 3.3 m/s with shoe and boot conditions. Temporal distance gait variables, ground reaction force components as well as lower limb joints angle, moment, and power were compared using Paired t-test and Statistical Parametric Mapping package for time-series analysis. Running with boots was associated with greater stride, step, flight, and swing times, greater flight length, and smaller cadence ($p<0.05$). The only effect of boots on lower limb joints kinematics during running was a reduction in ankle range of motion ($p<0.05$). Significantly greater hip flexor, abductor, and internal rotator moments, greater knee extensor and abductor moments, and ankle plantar flexor moments were observed at push-off phase of running as well as greater ankle dorsiflexor moment at early-stance in boot condition ($p<0.05$). Also, knee joint positive power was greater with a significant temporal shift in boot condition, suggesting a compensatory mechanism in response to limited ankle range of motion and the inability of the ankle joint to generate the required power. Our findings showed that running with boots is physically more demanding and is associated with a greater net contribution of muscles spanning hip and knee joints in order to generate more power and compensate for the ankle joint limitations, consequently, may increase the risk of both musculoskeletal injuries and degenerative joint diseases.

## Introduction

Lower limb injuries during running are very common in the active population with a 2-year incidence of 66% (73% for women and 62% for men) [1]. A total of 28 lower extremity alterations have been reported in the literature as injuries related to running, including medial tibial stress syndrome, Achilles tendinopathy, plantar fasciitis, patellar tendinopathy, ankle

**Competing interests:** The authors have declared that no competing interests exist.

sprain, Iliotibial band syndrome, hamstring tendinopathy, tibial stress fracture, and patellofemoral pain [2]. Overloading of the musculoskeletal structures resulting from repetitive microtraumas as well as biomechanical factors are the primary risk factors for running-related injuries [2, 3].

As the only interface between the foot and the ground, Footwear plays an important role in gait biomechanics and is of great concern in the management of lower limb injuries [4]. Amongst various types of footwear, boots are often used in rehabilitation, sports, and occupations to support and protect the foot and ankle on rough terrain and environmental conditions [5, 6]. The boots are usually characterized by less flexibility, poor cushioning, and more weight compared to other types of shoes [5] which might be associated with abnormal gait biomechanics and lower limb injuries. However, the boots-shaft height and stiffness restrict the free motions of the ankle and foot during walking, running, and stair climbing. This may in turn affect the related kinetic variables in gait. Consequently, further compensatory mechanisms may occur in the proximal joints and muscles load, leading to a higher risk of musculoskeletal injuries [6, 7]. Moreover, Sinclair et al. found that running with boots is associated with excessive patellofemoral joint load [8], Achilles tendon load [9], impact parameters [10], as well as greater foot eversion and tibial internal rotation [10] which are linked to the etiology of the related injuries.

The role of footwear on the incidence of running-related injuries has remained unknown. Despite an increasing number of studies on the influence of different footwear designs (e.g. maximalist, or minimalist) on gait mechanics in recreational runners, few studies have evaluated the biomechanical alterations during running with boots. Also, contradictory findings exist in the literature (Cikajlo and Matjačić (2007), Böhm and Hösl (2010)) which can partially be explained by the variations in the design and mechanical properties of boots between different studies [6, 7] and require more investigation.

Identifying the biomechanical alterations in lower limb joints while running with boots may help to better understand the underlying mechanisms of the related injuries. The results of this study may have clinical implications for treating people with footwear-related injuries who frequently use boots. Furthermore, this may aid to develop the footwear design in order to reduce the risk of both running and footwear-related injuries. Given that there are few studies in this field, therefore, the purpose of this study was to compare the temporal-distance gait variables, ground reaction forces (GRF), kinematics, and kinetics during running with boots compared with running shoes.

## Methods

### Participants

In this study, 17 healthy male university students (age: 23.06±2.58 years; body mass: 71.54 ±11.25 kg; height: 1.76±0.08 m) were studied. Participants were physically active and did not have any history of musculoskeletal abnormalities, neuromuscular deficits, bone fracture, major surgery, or any condition that affects normal running [11]. The protocol of the study was approved by the local research ethics committee (IR.BMSU.REC.1400.024) and written informed consent was obtained from each individual prior to the experiment.

### Experimental setup

Twenty-seven retro-reflective markers were placed on both right and left lateral wrist and elbow, acromion, superior iliac spines, knee epicondyles, ankle malleoli, calcaneus, and metatarsal heads. Additionally, a set of four non-collinear cluster markers mounted on a rigid plate was fixed on the lateral side of the thigh and shank segments using a wide elastic band [12] (for

details see S1 Table). A motion capture system (Miqus M3, Qualisys AB, Sweden) including eight cameras (200 Hz) and one Kistler force platform (type 9281EA, Kistler Instrumente AG, Switzerland) at 2000 Hz, were used to record both marker trajectories and GRF components, respectively, during the running task. Qualisys Track Manager software (QTM) (v. 2020.3, Qualisys AB, Sweden) software was used to operate both systems synchronously. A timing gate system was built to control the running velocity, using two Photocells at a distance of 1 m, before and after the center of the force platform.

The boots and running shoes used in this study were provided by a local manufacturer (three pairs from 41 to 43 EU sizes (Fig 1), Aghanezhad shoes Inc., Iran). Detailed information regarding the technical specifications of each one (for 42 EU size) can be found in Table 1.

## Protocol

The experimental design was a single-session crossover trial in which the participants were asked to perform running trials under wearing two footwear conditions (boots and shoes) along an 18-meter walkway. Running trials for each condition were repeated until at least four successful strides of the dominant leg were recorded. A trial was considered successful if the foot was landed at the middle of the force plate with a distinct heel strike and the running velocity of 3.3±5% m/s. The order of trials was counterbalanced with 10 min rest time between conditions to minimize the order and fatigue effects, respectively. A 3-second static position of the participant in each of the boot/shoe conditions was recorded. Prior to the data collection, the participants were allowed to have warm-up practices as long as they felt comfortable and familiarized with the boots/shoes conditions and the imposed running velocity.

## Data analysis

Marker trajectories were processed in QTM software including marker labeling, gap-filling, spike removal, time cropping, and C3D extraction. Virtual joint center markers were calculated for each static trial as follows: Hip joint center by regression equation defined by Harrington et al. (2007) [13] and knee, ankle, and forefoot joints center by averaging the medial and lateral epicondyle, malleolus, and metatarsal head markers, respectively. GRF data were low-pass filtered using zero-lag fourth-order Butterworth at a cut-off frequency of 50 Hz [14, 15].

A generic full-body model with a six degrees of freedom (DoF) joint between pelvis and ground, 3DoF hip joint, one DoF parametrized knee, and one DoF ankle and subtalar joints were used in this study [16]. The parametrized knee joint included knee adduction and internal rotation as well as vertical and anteroposterior joint translations as functions of knee flexion angle [17]. In order to be able to calculate knee moments in the frontal and transverse planes (Fig 2) while preserving the parametrized knee joint, the original model was modified similar to Saxby et al. (2016) and Lenton et al. (2018) [11, 18]. Also, all coordinates were unclamped to allow as much range of motion (RoM) as necessary (e.g. knee hyperextension).

Dynamic simulations carried out by OpenSim API (v.4.3., Stanford University, USA) [19]. All segments were linearly scaled based on the distance of experimental/virtual and model markers [20]. Following the scaling, the boots/shoes mass difference (0.3 kg) was appended to each foot segment (calcn_r/l) of scaled boots models. Within the Opensim workflow, the InverseKinematics and InverseDynamics tools were used to calculate the joints' kinematics and kinetics, respectively. Kinematic data were low-pass filtered with a 20 Hz cut-off frequency. Joint power was calculated by the product of joint angular velocity (in radians) and the moment [21].

The heel strike and toe-off events of the running events were detected using vertical foot velocity and maximum knee extension algorithms, respectively, which have been reported as the most

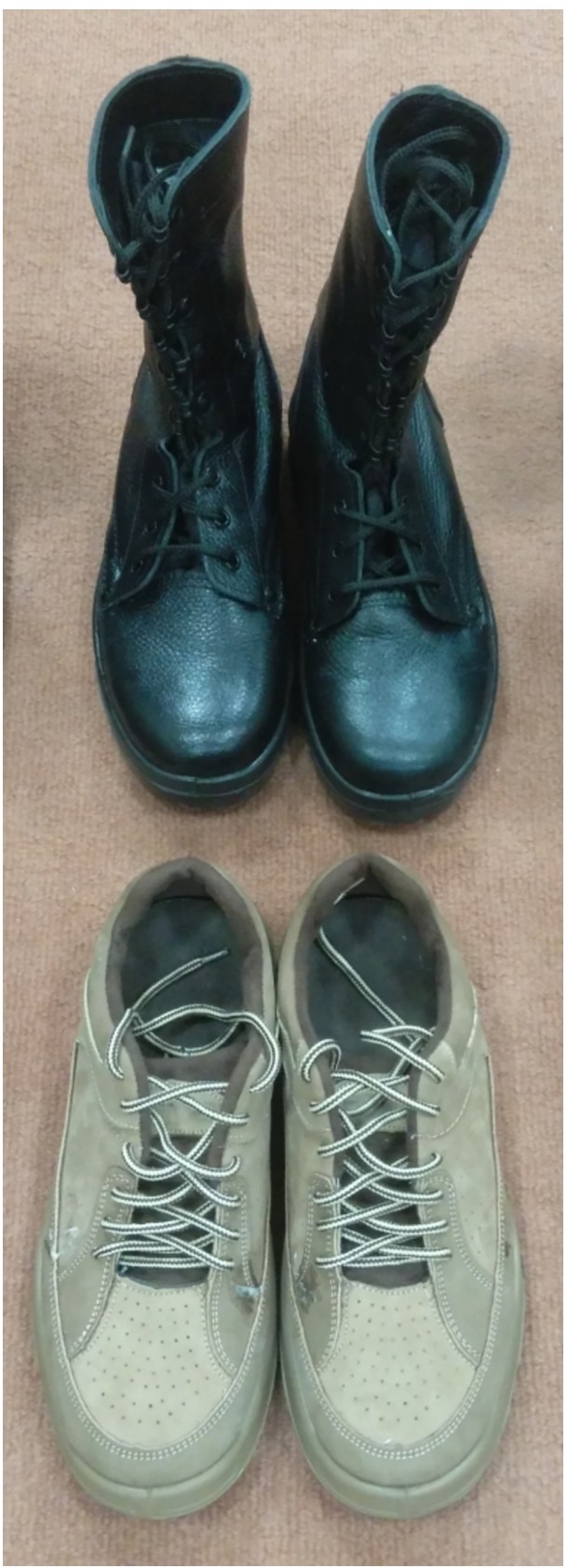

**Fig 1. Boots and shoes from 41 to 43 EU sizes.**

**Table 1. Technical specifications of boots and shoes.**

|  | Boot | Shoe |
|---|---|---|
| Weight (kg) | 1.228 | 0.627 |
| Shaft height (mm) | 222 | 89 |
| Heel thickness (mm) | 35 | 25 |
| Heel-to-toe drop (m) | 20 | 5 |
| Outsole material | Polyurethane—Rubber | Polyurethane |
| Outsole density (g/ml) | 0.45±0.05–1.15±0.15 | 0.45±0.05 |
| Outsole hardness | shoreA 50—shoreA 65 | shoreA 50 |
| Upper layer material | Cow leather | Nubuck leather |

accurate methods with the lowest errors amongst the other kinematic-based algorithms for detecting the running events (Fellin et al. (2010)) [22]. Gait velocity was determined using forward translation of pelvis mass center. Other temporal-distance gait variables including cadence, time of the stride, stance, step, swing, and flight, as well as the step width, step length, stride length, flight length, and foot clearance, were calculated using the foot marker trajectories.

The GRF, kinematics, and kinetics data were time-normalized on the scale of 0 to 100% corresponding to the two consecutive heel strikes for kinematic variables and the begin to the end of the stance phase of running for the kinetic variables. Ensemble average graphs of GRF vectors and kinetics were normalized to body weight and body mass, respectively, and the entire dataset (S1 File), as well as the data of all graphs including GRF (S2 Table), kinematics (S3 Table), joints moment (S4 Table), and joints power (S5 Table), are provided in the Supporting Information section. All computations were performed using a custom script (Python v.3.8, Python Software Foundation, USA).

## Statistical analysis

First, a Shapiro-Wilks test was conducted to examine the normality of the distributions of the data. All data were found to be normally distributed. Discrete temporal-distance gait variables of the boot and shoe conditions were compared using Paired t-tests. For a better understanding of any possible difference in kinematics and kinetics variables over the entire running cycle between the two footwear conditions, one-dimensional statistical parametric mapping was applied [23]. Significance level was set at $p < 0.05$ for all comparisons. All plots were generated using Matplotlib package [24].

## Results

### Temporal-distance gait variables

The temporal-distance gait variables during running with boots and running shoe conditions are reported in Table 2. The results indicate significantly less cadence (2.9%, $p = 0.04$) and greater step time (2.9%, $p = 0.03$), stride time (1.9%, $p = 0.05$), swing time (4%, $p = 0.008$), flight time (1.2%, $p = 0.005$), and flight length (5.7%, $p = 0.004$) were observed for boot condition. No significant difference existed for velocity, stance time, step width, step length, stride length, and foot clearance ($p > 0.05$). The results also show that the running velocity was successfully controlled during data collection as the mean values were kept in the range of 3.3±5% m/s.

### Ground reaction force

The anterior-posterior and medial-lateral components of the GRF in running with boots were not statistically different from those of running with shoes (Fig 3). However, the vertical GRF

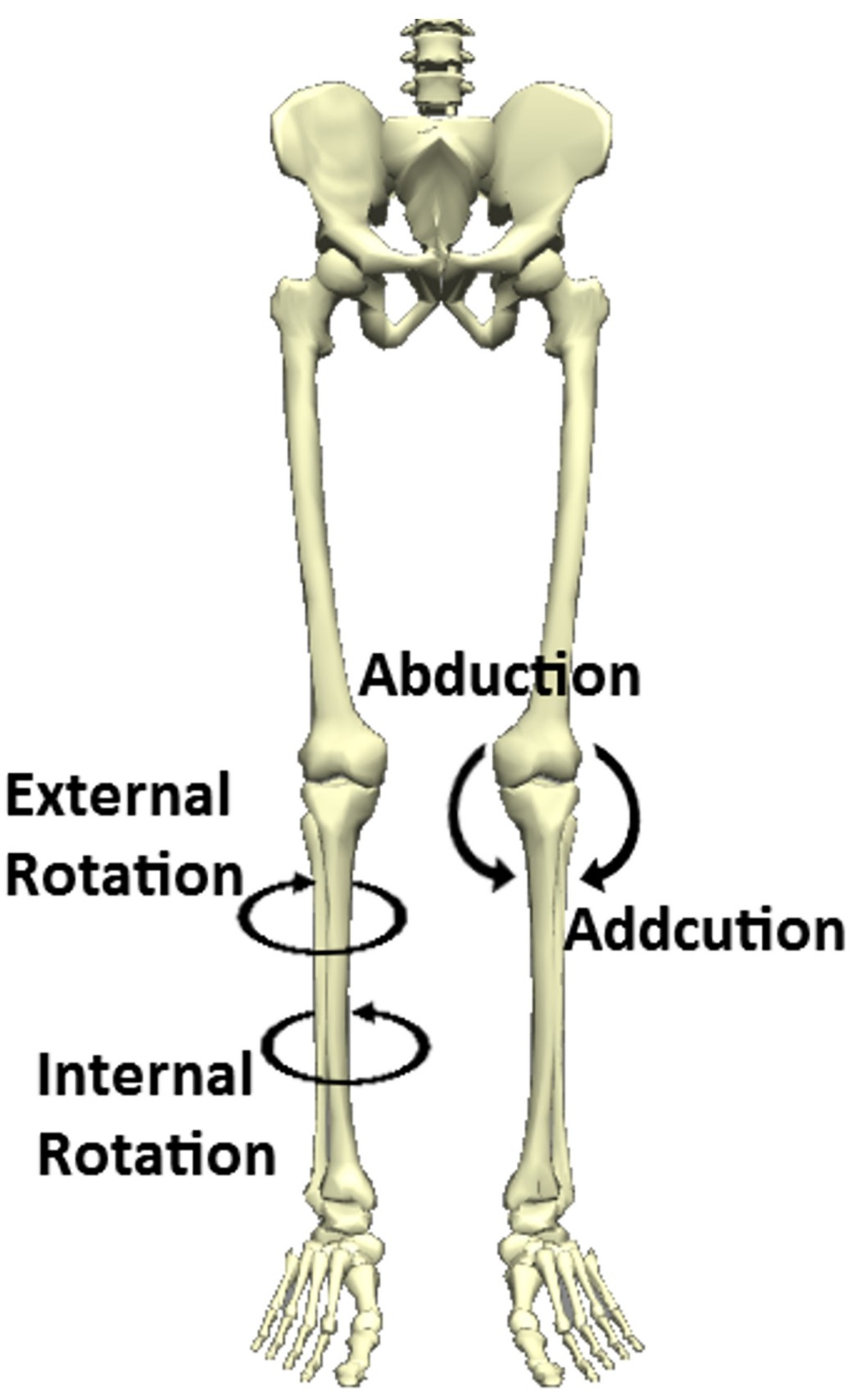

**Fig 2. Knee motions in the frontal and transverse planes.**

**Table 2. Temporal-distance gait variables for boot and shoe conditions.**

|  | Boot (mean±SD) | Shoe (mean±SD) | t (*p*-value) |
|---|---|---|---|
| Velocity (m/s) | 3.28 ± 0.09 | 3.31 ± 0.07 | -1.33 (0.2) |
| Cadence (step/min) | 157.47 ± 9.07 | 162.17 ± 10.19 | -2.23 (0.04) |
| Stride time (ms) | 763.38 ± 37.55 | 749.19 ± 38.82 | 2.11 (0.05) |
| Stance time (ms) | 269.4 ± 13.46 | 273.97 ± 17.63 | -1.5 (0.15) |
| Step time (ms) | 382.28 ± 21.92 | 371.47 ± 23.76 | 2.29 (0.03) |
| Swing time (ms) | 492.94 ± 37.12 | 474.04 ± 32.61 | 3.03 (0.008) |
| Flight time (ms) | 113.24 ± 21.48 | 97.5 ± 17.41 | 3.23 (0.005) |
| Stride length (cm) | 247.87 ± 10.35 | 245.39 ± 10.46 | 1.3 (0.21) |
| Step width (cm) | 8.03 ± 2.35 | 8.46 ± 2.66 | -0.66 (0.51) |
| Step length (cm) | 112.24 ± 7.64 | 114.88 ± 7.98 | -1.69 (0.11) |
| Flight length (cm) | 105.75 ± 5.8 | 100.11 ± 5.97 | 3.33 (0.004) |
| Foot clearance (cm) | 25.64 ± 6.72 | 25.06 ± 5.37 | 0.42 (0.68) |

component in running with boots was significantly greater in the weight acceptance (4.8–7.4% of stance, *p* = 0.3) and mid-stance (57–66.7% of stance, *p*<0.001) phases compared with shoes.

## Kinematics

In running with the boots, the ankle plantarflexion at heel strike phase (1.9–5.4% of stride, *p* = 0.04) was greater, but, the ankle dorsiflexion at the mid-stance (19.2–25.4% of stride, *p* = 0.019), and the ankle plantarflexion at initial swing (37.3–55.1% of stride, *p*<0.001) phases were significantly smaller than those in running shoes (Fig 4). The RoM of the hip, knee, and subtalar joints in running with the boots was not statistically different compared with shoes.

## Kinetics

In running with boots, the hip flexor moment in late-stance (92.1–100% of stance, *p*<0.001) and hip abductor moment (57.4–78.2% of stance, *p*<0.001), hip internal rotator moment (50.2–69.1% of stance, *p*<0.001), knee extensor moment (43.1–71% of stance, *p*<0.001), and knee abductor moment (45.6–73.9% of stance, *p*<0.001) in the mid to late-stance phases as well as ankle dorsiflexor moment in early stance (0–13.1% of stance, *p* = 0.002) and ankle plan-tarflexor moment in mid-stance (63–76.5% of stance, *p* = 0.002) and late stance (94.7–100% of

### Ground Reaction Force

**Fig 3. Ground reaction force components for boot and shoe conditions.**

## Joint Kinematics

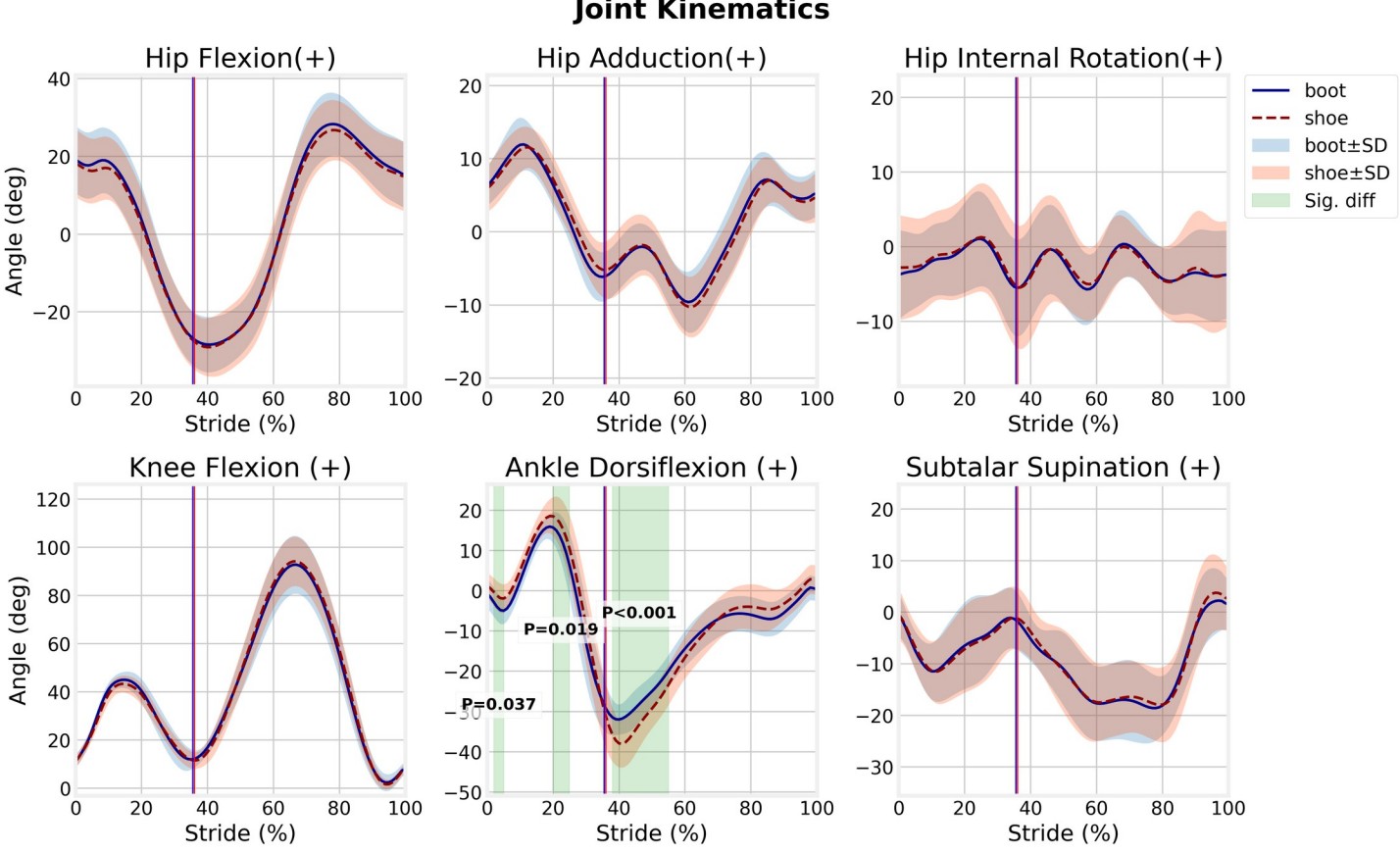

**Fig 4. Lower extremity joints kinematics for boot and shoe conditions.**

stance, $p = 0.03$) were significantly greater than those in running with shoes (Fig 5). The knee internal rotator moment and moment in the subtalar joint were not statistically different compared with running shoes.

The hip sagittal power absorption in late-stance at 94–100% of the stance ($p = 0.001$) and hip frontal power generation at 70.7–78.9% of the stance ($p<0.001$) were greater in boot condition (Fig 6). The ankle power generation at 54.4–57.1% of the stance phase was lower in the boots ($p = 0.032$). Also, in running with the boots, the knee joint displayed greater power generation with a significant temporal shift at 35.6–45.1% ($p<0.001$) and 55.7–71.9% ($p<0.001$) of the stance phase compared with running shoes.

## Discussion

The aim of the current study was to compare the gait characteristics in over-ground running with boots with those of running with shoes. The temporal-distance gait variables, as well as the lower limb joints angle, moment, and power, were quantified and compared during the two running conditions. To eliminate the influence of running speed, all subjects were trained to run at about 3.3 m/s.

The results of the present study showed that the step rate was significantly smaller in running with boots than in running with shoes. It has been reported that reduced running cadence increases the applied loads on the patellofemoral joint, which is associated with a higher risk of patellofemoral pain [25] as well as tibial stress fracture [15]. To reach the desired velocity, the

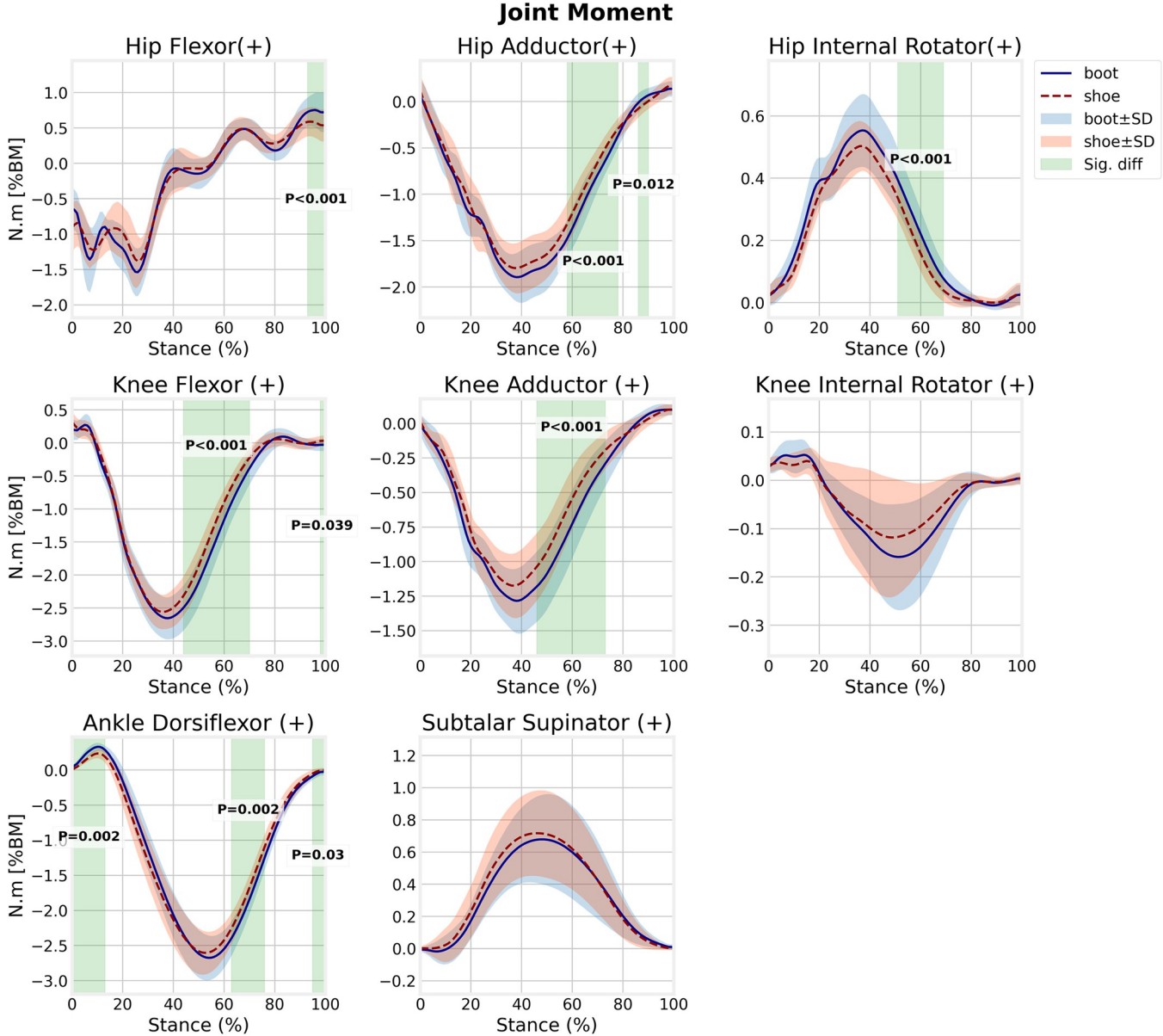

**Fig 5. Lower extremity joints moment for boot and shoe conditions.**

decreased stance time, together with the increased swing time, flight time, and length in boot condition, shows that running with boots demands more physical effort than in shoe condition. No significant differences were observed on other temporal measures in running under the two conditions.

The present study also showed that the magnitude of the peak vertical GRF was higher in boot condition which is in agreement with previous reports [26–28]. This increased peak vertical GRF can partly be explained by the greater mass of the boots [29]. It is postulated that the higher mass of the boots may increase the effective mass of the foot, and consequently the rotational inertia of the leg [30]. This may result in more muscle load, fatigue rate, energy expenditure, and consequently, increase the risk of injury [31]. Visual inspection of vertical GRF (Fig 3) also shows decreased time to impact peak in boot condition which is not surprising for the

## Joint Power

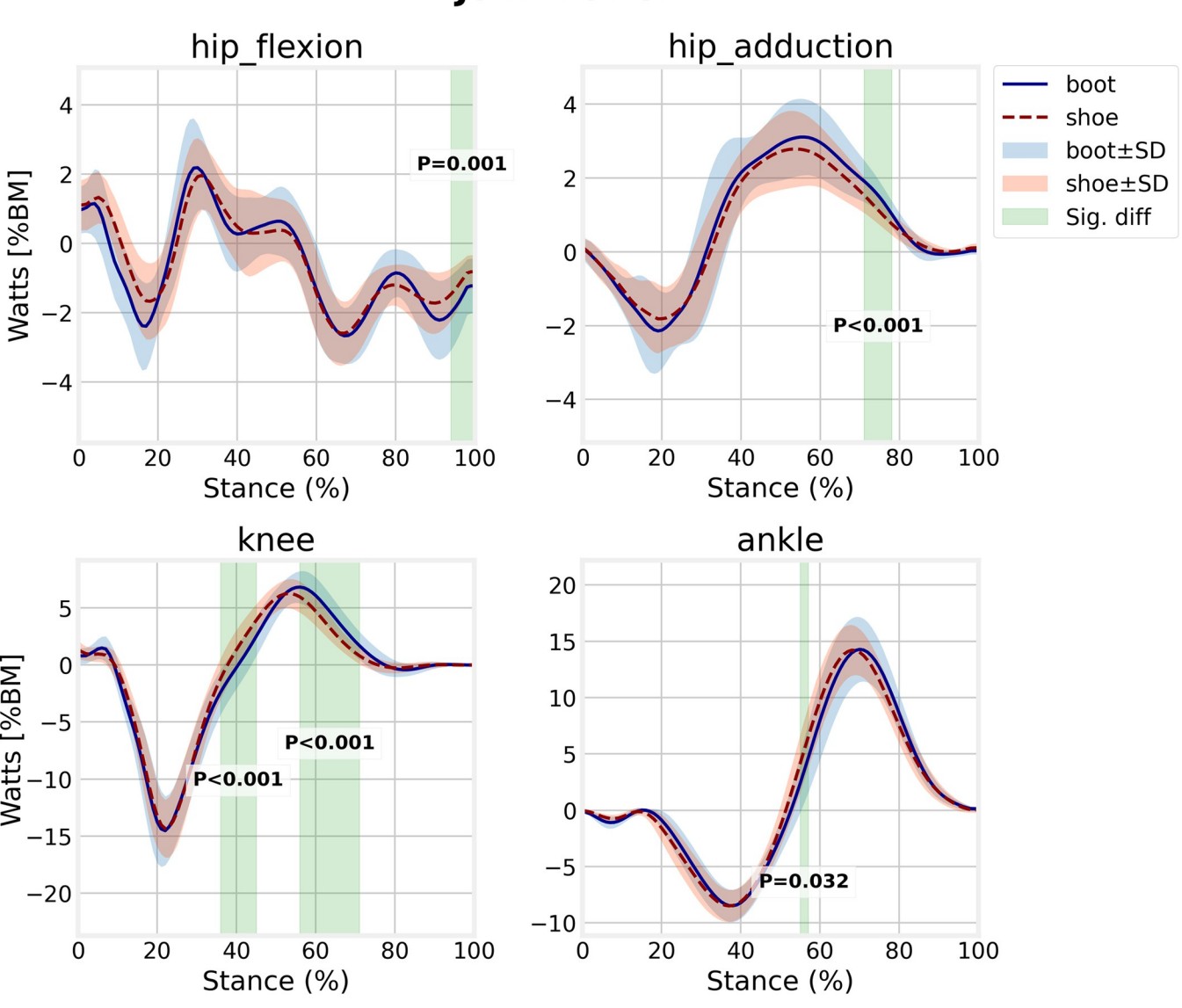

**Fig 6. Lower extremity joints power for boot and shoe conditions.**

less cushioned footwear and it is in agreement with previous studies [10]. Literature found that the time to peak is more important than the magnitude of the peak itself in running injuries [14, 32]. Greater variability in the medial-lateral component of GRF existed that has been noted previously [26] and there were also non-significant less breaking and propulsion peaks of the anterior-posterior component of GRF in the boot condition.

In running with the boots the RoM of the ankle joint in entire stride time, particularly at the mid-stance and initial swing phases was decreased which is in agreement with findings of the previous studies [6, 7, 10]. Additionally, increased ankle plantarflexion in early-stance in the boot condition supports the results obtained by Sinclair and Taylor (2014) [10]. This makes sense for the footwear with a stiff outsole and 15 mm greater heel-to-toe drop (Table 1). In the present study, there was not any significant difference between the boot and shoe conditions on the RoM of the hip, knee, and subtalar joints. This finding is similar to walking with

hard and soft boots, investigated by Böhm and Hösl (2010) [6] and Cikajlo and Matjačić (2007) [7]. In contrast, Sinclair and Taylor (2014), reported significantly greater knee flexion and ankle eversion at heel strike in boot condition [10].

We observed a significantly greater ankle dorsiflexor moment in the early stance in the boot condition. At heel strike, eccentric contraction of the muscles in the anterior compartment of the leg controls the rapid ankle plantarflexion. This increase is not surprising for stiff and heavy footwear like boots. Also, ankle plantarflexor moment was greater for boot condition compared with shoes (significant in mid to late-stance) and this is in agreement with Sinclair et al. (2015) [9] and in contrast with Cikajlo and Matjačić (2007) for walking with hard and soft boots [7]. The knee moments in all three axes were greater in the boot condition compared with the shoe condition, whereas Sinclair et al. (2015) found greater knee abductor moment in running-trainer shoes compared with the boots [8]. Increased knee extensor and abductor moments during running with boots are associated with increased knee medial compartment loading [33] which in turn, could be a risk factor for early onset and progression of knee osteoarthritis (OA) [34]. Furthermore, excessive knee abductor moment has been found as a predictor of knee anterior cruciate ligament (ACL) injury [35]. Greater hip flexor, abductor, and internal rotator moments were found for the boot condition in mid to late-stance phases which have not been previously reported. Studies have found cumulative hip moment, particularly in the frontal plane is associated with subsequent radiographic progression of hip OA [36]. Therefore, running with boots increases the risk for degenerative joint diseases in the long term.

Joint power analysis revealed a significant difference for positive ankle power between conditions. However, visual inspection of the average ensemble graphs (Fig 6) indicates that this difference is not on the magnitude, but a temporal shift for boot condition that delays ankle power generation. This finding is partially in contrast with Cikajlo and Matjačić (2007) who reported larger ankle power generation for softer boots during walking [7] and Böhm and Hösl (2010) who didn't observe any difference in ankle positive power during walking with hard and soft boots [6]. Also, the magnitude of the knee joint positive power was greater in the boot condition in addition to a significant delay in power generation. In contrast with Böhm and Hösl (2010) [6], we did not observe any difference in the knee and ankle negative power. Hip joint power in the frontal plane was significantly increased in mid to late-stance for boot condition and in association with increased hip abductor moment, suggesting the need for more pelvis stabilization in the push-off phase. Excessive hip, knee, and ankle joint moments and powers during running with the boots suggest a greater net contribution of muscles spanning each joint compared with the shoe condition which may lead to more load on soft tissue and joint, muscle fatigue, and energy expenditure during gait which in turn may increase the risk of both musculoskeletal injuries and degenerative joint diseases in long term.

The differences in magnitude and pattern of the hip frontal plane, knee, and ankle power generation may be a mechanical adaptation in response to the limited ankle RoM, mainly due to the specific characteristics of the boots such as shaft height and the stiffness of the upper layer, vamp, and outsole [26]. Moreover, excessive time and length of flight phase found for boot condition (Table 2) require more propulsive energy at the push-off phase of gait. When the ankle joint is not allowed to be plantarflexed sufficiently, it cannot generate adequate power for propulsion [7], and therefore, the proximal joints must compensate for this limitation. Particularly, excessive work is done by knee extensor muscles to provide more energy to accelerate the limb toward swing phase. This finding has not been well addressed in boot-related studies.

Theisen et al. (2016) in their prospective study have shown limited evidence for the role of footwear in running-related injuries [37]. Several investigations suggested the differences

found between conditions are not related to footwear itself, but the transition to the new footwear [38]. Nonetheless, caution must be taken when interpreting and implementing the results. It is however important to mention that our findings may be important in terms of understanding the mechanisms of musculoskeletal injuries and degenerative joint diseases derived from abnormal gait biomechanics. Additionally, the alterations that occurred in lower limb joint biomechanics during running with boots may be clinically significant for people who frequently use boots for occupation, sport, or rehabilitation. Long-term usage of boots may increase the risk of ACL injury as well as hip and knee OA. More prospective and randomized controlled trials must be conducted to understand the long-term effects of wearing boots on gait biomechanics and its contribution to the high incidence of musculoskeletal injuries and degenerative joint diseases in boots users.

The discrepancies between studies might be explained in part by the differences in the applied methods. Computing joint angles using skin-mounted markers is subject to soft tissue artifact [39]. The skeletal model we used for dynamic simulation was constrained, meaning that not all translational and rotational coordinates were allowed. This approach has several advantages compared to unconstrained inverse kinematics (6DoF model), including consistent segment length as well as a lack of non-physiological joint translations such as dislocation and impenetration [40]. In terms of the knee motion in the frontal and transverse planes, Benoit et al. (2005) found notable observational errors between skin-mounted and bone-pin markers during walking and cutting maneuvers [41]. Therefore, these coordinates had been constrained as a function of the knee flexion in the original model [16, 17]. The other advantage of this model is the physiological axes of rotation for knee, ankle, and subtalar joints [16].

Attempts were made to conduct comprehensive analyses in the biomechanics of running by using time-series analysis and including more gait variables. However, it is still not possible to determine which specific characteristic of the boots caused each of these biomechanical alterations. One advantage of this investigation is the greater number of participants compared with similar studies [8–10] which were less than 14 subjects. A possible limitation in this study is that the foot markers were mounted on footwear rather than foot skin. Consequently, the measured subtalar kinematics may not represent the foot movements within the shoes/boots. Sinclair et al. (2013) stated that the current method is the most appropriate one since cutting a hole on the footwear surface as a possible alternative approach affects the structural integrity of the footwear as well as the runners' perception [42]. Lack of electromyography and energy expenditure data is another limitation that could have helped us in the interpretation as well as validation of our findings.

## Conclusion

The present study indicates that running at an identical velocity with boots is physically more demanding than running with shoes and requires a further contribution of lower limb muscles in order to generate more energy for propulsion. Running with boots is associated with increased hip and knee joints moment imposing a greater load on joints surface which might be suggestive of hip and knee OA as well as knee ACL injury. Mechanical adaptations resulting from the boot itself can put the individual at greater risk for musculoskeletal injuries and degenerative joint diseases. Future studies and interventions must be conducted on the design and materials of the boots to provide clinical benefits to users.

## Supporting information

**S1 File. DataSet.** The file contains experimental markers and force data (TRC and MOT), OpenSim model (OSIM), setup (XML), and output (STO) files for 17 participants, 2

conditions (boot\shoe), and 4 trials which are stored in separate folders.
(ZIP)

**S1 Table. Marker set descriptions.**
(DOCX)

**S2 Table. Data of GRF graphs.** Data (Mean±SD) of each participant and each component of
GRF.
(XLSX)

**S3 Table. Data of kinematics graphs.** Data (Mean±SD) of each participant and each joint.
(XLSX)

**S4 Table. Data of joints moment graphs.** Data (Mean±SD) of each participant and each
joint.
(XLSX)

**S5 Table. Data of joints power graphs.** Data (Mean±SD) of each participant and each joint.
(XLSX)

## Acknowledgments

The authors appreciate all volunteers who kindly participated in this study.

## Author Contributions

**Conceptualization:** Alireza Shamsoddini, Mohammad Taghi Hollisaz.

**Data curation:** Alireza Shamsoddini, Mohammad Taghi Hollisaz.

**Formal analysis:** Alireza Shamsoddini, Mohammad Taghi Hollisaz.

**Funding acquisition:** Alireza Shamsoddini, Mohammad Taghi Hollisaz.

**Investigation:** Alireza Shamsoddini, Mohammad Taghi Hollisaz.

**Methodology:** Alireza Shamsoddini, Mohammad Taghi Hollisaz.

**Project administration:** Alireza Shamsoddini, Mohammad Taghi Hollisaz.

**Resources:** Alireza Shamsoddini, Mohammad Taghi Hollisaz.

**Software:** Alireza Shamsoddini, Mohammad Taghi Hollisaz.

**Supervision:** Alireza Shamsoddini, Mohammad Taghi Hollisaz.

**Validation:** Alireza Shamsoddini, Mohammad Taghi Hollisaz.

**Visualization:** Alireza Shamsoddini, Mohammad Taghi Hollisaz.

**Writing – original draft:** Alireza Shamsoddini, Mohammad Taghi Hollisaz.

**Writing – review & editing:** Alireza Shamsoddini, Mohammad Taghi Hollisaz.

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
