## [Decision Letter · Decision Letter 0]

18 Apr 2022

PONE-D-22-07506Biomechanics of running: a special reference to the comparisons of wearing boots and shoesPLOS ONE

Dear Dr. Shamsoddini,

Thank you for submitting your manuscript to PLOS ONE. After careful consideration, we feel that it has merit but does not fully meet PLOS ONE’s publication criteria as it currently stands. Therefore, we invite you to submit a revised version of the manuscript that addresses the points raised during the review process.

We look forward to receiving your revised manuscript.

Kind regards,

Osama Farouk

Academic Editor

PLOS ONE

Journal Requirements:

Reviewers' comments:

Reviewer's Responses to Questions

**Comments to the Author**

1. Is the manuscript technically sound, and do the data support the conclusions?

Reviewer #1: Yes

Reviewer #2: Yes

Reviewer #3: Yes

2. Has the statistical analysis been performed appropriately and rigorously? 

Reviewer #1: Yes

Reviewer #2: N/A

Reviewer #3: Yes

3. Have the authors made all data underlying the findings in their manuscript fully available?

Reviewer #1: No

Reviewer #2: No

Reviewer #3: Yes

4. Is the manuscript presented in an intelligible fashion and written in standard English?

Reviewer #1: Yes

Reviewer #2: Yes

Reviewer #3: No

5. Review Comments to the Author

Reviewer #1: This manuscript examines the effects of shoes and boots on running biomechanics. The manuscript is well designed, written, and suitable for publication after peer review. I suggest the following:

1) L22-23: "Reduced ankle range of motion was the only kinematic effect of running with boots"? This is not clear; please rephrase.

2) L35-36: "Lower limb injuries during running are very common in the active population with an annual incidence of between 20 to 80% [1]. " the presented percentage of injuries linked to running activity is a bit general and somehow unrealistic in the upper bound (80%). Could you please be more specific and clear on this point, especially how can 80% of runners may get injured? This is a very important point since some readers/researchers may use this paper later as a reference.

3) L36: "A total of 28 lower extremity conditions have been reported in the literature as injuries...". Could you please replace conditions with alterations?

4) By the end of the introduction, it is better to emphasize the importance of the scientific question that you are trying to answer via this study. In other words, what is the added value of the proposed study?

5) It's important to add a picture of the used shoes and boots. This may help the readers to draw some additional conclusions.

6) Here is not clear if the authors have used different sizes of the boots/shoe or just one size (42 EU), as indicated in Table 1. An explanation needs to be addressed if it is one size.

7) The authors need to present the adduction-abduction and internet-external knee rotations, which are additional tools for the reader to draw some observations on joint load distributions and muscle activations.

8) Here not clear why the authors have ignored muscles activation measurements via EMG or even theoretical optimization via OpenSim; this info may endorse a lot of presented observations in the discussion section.

9) Paper needs to go through minor language editing to avoid some grammatical errors. Herein are some of them were detected:

L199: a comma before which

L207: partly be explained

L216: in boot conditions

L243: positive ankle

L244: average ensemble

L252: the frontal

L283: a lack

Reviewer #2: Very goid title to be explored because there is not too much article about this title

Please provide all data in figures

The way you analyses the data And complete the methods and results section

Statistical analysis in not what we expected from this type of study

Reviewer #3: Title: Specify if the shoes are running or everyday shoes.

Abstract. Explain what means a greater net contribution (line 30).

Introduction. Correct: patellar tendinopathy (not Patellar) (line 38).

Rephrase the first two sentences of the second paragraph, as the grammar is not very accurate.

Replace Sinclair and his colleagues with Sinclair et al.

Name the studies the authors refer to (line 62).

Discussion. For the referring studies, add the year of publication in brackets (for example, Sinclair (2014)).

Explain what boots literature means.

Name the prospective studies the authors refer to (line 269).

Specify the difference between the number of the participants of the current study and the similar ones.

Point out the clinical implications of the current study.

6. PLOS authors have the option to publish the peer review history of their article (what does this mean?). If published, this will include your full peer review and any attached files.

Reviewer #1: No

Reviewer #2: No

Reviewer #3: No

---

## [Author Response · Author response to Decision Letter 0]

13 May 2022

Responses to the comments of the editor and reviewers are provided in the "Response to Reviewers.docx" file.

---

## [Decision Letter · Decision Letter 1]

13 Jun 2022

Biomechanics of running: a special reference to the comparisons of wearing boots and running shoes

PONE-D-22-07506R1

Dear Dr. Shamsoddini,

We’re pleased to inform you that your manuscript has been judged scientifically suitable for publication and will be formally accepted for publication once it meets all outstanding technical requirements.

Kind regards,

Osama Farouk

Academic Editor

PLOS ONE

Additional Editor Comments (optional):

Reviewers' comments:

Reviewer's Responses to Questions

**Comments to the Author**

1. If the authors have adequately addressed your comments raised in a previous round of review and you feel that this manuscript is now acceptable for publication, you may indicate that here to bypass the “Comments to the Author” section, enter your conflict of interest statement in the “Confidential to Editor” section, and submit your "Accept" recommendation.

Reviewer #1: All comments have been addressed

Reviewer #3: All comments have been addressed

2. Is the manuscript technically sound, and do the data support the conclusions?

Reviewer #1: Yes

Reviewer #3: Yes

3. Has the statistical analysis been performed appropriately and rigorously? 

Reviewer #1: Yes

Reviewer #3: Yes

4. Have the authors made all data underlying the findings in their manuscript fully available?

Reviewer #1: Yes

Reviewer #3: Yes

5. Is the manuscript presented in an intelligible fashion and written in standard English?

Reviewer #1: Yes

Reviewer #3: Yes

6. Review Comments to the Author

Reviewer #1: The authors have carefully answered the reviewers' comments and edited the paper accordingly. Therefore, on my side, I have nothing to add; I suggest the acceptance of the proposed manuscript.

Reviewer #3: All my comments were answered. The authors have addressed adequately all my comments.

The article brings important data related to the biomechanics of running.

7. PLOS authors have the option to publish the peer review history of their article (what does this mean?). If published, this will include your full peer review and any attached files.

Reviewer #1: No

Reviewer #3: No

---

## [Editor Report · Acceptance letter]

15 Jun 2022

PONE-D-22-07506R1 

Biomechanics of running: a special reference to the comparisons of wearing boots and running shoes 

Dear Dr. Shamsoddini:

I'm pleased to inform you that your manuscript has been deemed suitable for publication in PLOS ONE. Congratulations! Your manuscript is now with our production department. 

Kind regards, 

on behalf of

Dr. Osama Farouk 

Academic Editor

PLOS ONE